# Polyomavirus Wakes Up and Chooses Neurovirulence

**DOI:** 10.3390/v15102112

**Published:** 2023-10-18

**Authors:** Arrienne B. Butic, Samantha A. Spencer, Shareef K. Shaheen, Aron E. Lukacher

**Affiliations:** Department of Microbiology and Immunology, Penn State College of Medicine, Hershey, PA 17033, USA; abb6377@psu.edu (A.B.B.); sas7639@psu.edu (S.A.S.); shareef.k.shaheen@psu.edu (S.K.S.)

**Keywords:** JCPyV, MuPyV, polyomavirus, progressive multifocal leukoencephalopathy (PML), CNS, antiviral immunity, neurotropic virus

## Abstract

JC polyomavirus (JCPyV) is a human-specific polyomavirus that establishes a silent lifelong infection in multiple peripheral organs, predominantly those of the urinary tract, of immunocompetent individuals. In immunocompromised settings, however, JCPyV can infiltrate the central nervous system (CNS), where it causes several encephalopathies of high morbidity and mortality. JCPyV-induced progressive multifocal leukoencephalopathy (PML), a devastating demyelinating brain disease, was an AIDS-defining illness before antiretroviral therapy that has “reemerged” as a complication of immunomodulating and chemotherapeutic agents. No effective anti-polyomavirus therapeutics are currently available. How depressed immune status sets the stage for JCPyV resurgence in the urinary tract, how the virus evades pre-existing antiviral antibodies to become viremic, and where/how it enters the CNS are incompletely understood. Addressing these questions requires a tractable animal model of JCPyV CNS infection. Although no animal model can replicate all aspects of any human disease, mouse polyomavirus (MuPyV) in mice and JCPyV in humans share key features of peripheral and CNS infection and antiviral immunity. In this review, we discuss the evidence suggesting how JCPyV migrates from the periphery to the CNS, innate and adaptive immune responses to polyomavirus infection, and how the MuPyV-mouse model provides insights into the pathogenesis of JCPyV CNS disease.

## 1. Introduction

JCPyV is a member of the family *Polyomaviridae*, a group of non-enveloped double-stranded DNA viruses that are highly host-specific and are found in a wide range of species [1]. One of fourteen human-specific polyomaviruses, JCPyV is notable due to its pathogenic neurotropism. The virus can cause a multitude of diseases in the CNS, such as JCPyV meningitis, JCPyV granule cell neuronopathy, and, most prominently, PML, a demyelinating disease involving lytic infection of glial cells [2,3,4]. These conditions are relatively rare, despite the ubiquitous presence of JCPyV in humans worldwide. Indeed, seroprevalence for JCPyV ranges from 40 to 80 percent and increases with age, with initial exposure likely occurring in childhood [5,6,7,8]. The transmission of JCPyV presumably occurs via the fecal–oral route, as viral particles have been found in the urine and stool [9,10]. Moreover, JCPyV is a frequent contaminant of urban sewage [11]. Transmission via the respiratory route is also possible as JCPyV has been found in the tonsillar tissue of various populations [12,13].

In an immunocompetent individual, JCPyV infection is typically asymptomatic. The virus can establish several sites of infection in tissues such as the kidney, bone marrow, and lymphoid tissue [3]. Viral infiltration of the CNS occurs in an immunocompromised setting, which allows the virus to leave peripheral sites of infection and travel to the CNS. Where and how JCPyV infiltrates the brain is unknown and is challenging to explore due to the historical lack of an easily experimentally manipulatable small animal model. Such a model would provide important insight into aspects such as early viral pathogenesis—prior to manifestation of neurological symptoms—and immune mechanisms associated with the control of JCPyV.

Mouse polyomavirus (MuPyV), the inaugural polyomavirus, has been established as a model for studying CNS-associated polyomavirus infection [14]. MuPyV is prevalent in wild mouse populations [15] and, like JCPyV, causes an asymptomatic and persistent infection in its immunocompetent host [16,17]. This review explores the outstanding question of how JCPyV may infiltrate the CNS from the periphery as implicated by findings employing the MuPyV model. We will cover peripheral polyomavirus persistence, how the virus resurges in peripheral sites, escapes antiviral immunity, and invades the CNS, which CNS-resident and infiltrating cells the virus infects, and the immunological control of polyomaviruses in the CNS.

## 2. Structure and Lifecycle of Polyomaviruses: JCPyV and MuPyV

JCPyV and MuPyV are both classified as members of the mammalian polyomavirus genus, *Orthopolyomavirus*, based on nucleotide sequences of large T antigen (LTAg) and structural proteins such as VP1 [18]. Although phylogenetic analyses (which are usually based on sequence homology of either LTAg or VP1) show that, compared to JCPyV, MuPyV is more closely related to Merkel cell polyomavirus, which is another human-specific polyomavirus, the two viruses’ similarities in the viral genome and lifecycle are more relevant in showing how MuPyV can model JCPyV pathogenesis [18,19,20].

### 2.1. Viral Structure and Genome

Both JCPyV and MuPyV have nonenveloped icosahedral T = 7 capsids encasing a covalently closed circular genome packaged into a supercoiled minichromosome. Both genomes are similar in size, with JCPyV being 5130 base pairs long and MuPyV being 5300 base pairs long [21,22]. As shown in Figure 1, the genomes of both viruses are divided into an early region and a late region based on the temporal expression of proteins during their replication cycle. For JCPyV, the early region encodes two nonstructural proteins, LTAg, and small T antigen (STAg), and three T antigen splice variants, T’135, T’136, and T’165 [5,23]. The late region comprises three structural capsid proteins—VP1, VP2, and VP3—as well as an agnoprotein and two microRNAs (miRNAs) that are processed from a pre-miRNA designated pre-miRJ-1 [21,24]. Because the miRNAs are perfectly complimentary to LTAg, they regulate the stability of LTAg transcripts [24]. MuPyV also encodes for a precursor miRNA that is processed into two mature miRNAs, which coordinate the cleavage of early mRNA transcripts [24]. The capsids of both JCPyV and MuPyV are organized into 72 pentamers of VP1, and each pentamer is associated with either VP2 or VP3 [25]. The early and late regions are separated by a non-coding control region (NCCR), which contains the origin of replication and binding sites for host proteins that determine species and cell tropism [26]. While the common nonpathogenic strain, referred to as the archetype or wild-type (WT) strain, contains a canonically organized NCCR, pathogenic variations of JCPyV found in the brain display extensively rearranged NCCRs, indicating that such rearrangements facilitate neurotropism [27,28,29,30].

Likewise, the genome of MuPyV is divided into an early and late region separated by an NCCR, with the early region encompassing STAg and LTAg and the late region comprising the structural proteins VP1, VP2, and VP3 [22]. Unlike JCPyV, MuPyV lacks an agnoprotein and encodes for a middle T antigen (MTAg), a membrane-localized oncoprotein that behaves as a constitutively active growth factor receptor [31,32]. Figure 1 depicts the similarities and differences between the MuPyV and JCPyV genomes.

### 2.2. Viral Life Cycle

Attachment and entry of both MuPyV and JCPyV are possible via receptors containing sialic acid residues. In the case of JCPyV, the attachment involves binding to the α2,6-linked sialic acid-containing lactoseries tetrasaccharide c (LSTc) receptor motif, though certain isolates of JCPyV have been found to bind sialic acid-containing ganglioside receptors [33]. Attachment to this receptor motif occurs via VP1, the secondary structure of which is composed of antiparallel β strands connected by loops that contain residues critical for receptor binding [34]. Viral entry is then assisted by interaction with 5-hydroxytryptamine 2 (5-HT_2_) receptors, which bind serotonin [35]. Similarly, MuPyV binds ganglioside receptors containing terminal sialic acid residues. Two such receptors have been identified as GD1a and GT1b, and viral entry is facilitated by the α4β1 integrin [36,37]. Interestingly, the immunohistochemistry of C57BL/6 mouse brains has shown that.

GT1b and GD1a are both expressed highly in neuronal tissue, with the former being expressed in more specific parts of the brain, such as the cerebral cortex and hippocampus, and the latter being more broadly expressed in both white and gray matter [38]. The distribution of these gangliosides supports the capability of MuPyV to infect the brain tissue of its host.

JCPyV and MuPyV differ in internalization processes (Figure 1). While MuPyV is internalized via caveolin-mediated endocytosis, JCPyV is internalized via clathrin-mediated endocytosis [39]. Recent work suggests that JCPyV may also be internalized via extracellular vesicles (EVs), which are cell membrane-derived vesicles that provide a means of intercellular communication and whose structure and contents are heterogeneous [40,41].

After internalization, both JCPyV and MuPyV transition to Rab5^+^ early endosomes, but whereas MuPyV moves to Rab11^+^ endosomes, a transition that appears to be dependent on the acidic endosomal environment, JCPyV enters a caveolin-1^+^ late endosome [42,43]. Both viruses localize to the endoplasmic reticulum (ER) and utilize the ER-associated degradation pathway (ERAD) to enter the cytosol. Throughout this process, partial degradation of the viral capsid occurs. Much of what is known regarding the uncoating of polyomaviruses within the ER is from studies of MuPyV and simian virus 40 (SV40), a well-studied polyomavirus that is naturally found in rhesus monkeys but can be transmitted to humans [44,45]. The architecture of MuPyV and SV40 is partly stabilized by disulfide bonds as well as calcium ions [44]. Both viruses interact with members of the protein disulfide isomerase family, resulting in the disruption of the aforementioned disulfide bonds [44,46]. Uncoating also involves the dissociation of calcium ions from the capsid, which, in the case of SV40, may be expedited by lower calcium concentration in the cytosol than the ER [47,48]. Uncoating of JCPyV may also be dependent on calcium, as thapsigargin, which blocks calcium transport into the ER, reduces in vitro infection of a JCPyV-based pseudovirus [49]. Both JCPyV and MuPyV ultimately end up in the nucleus, where the viral genome is released, and transcription of the early region occurs, resulting in the expression of the T antigens specific to each virus [50,51].

The functions of both LTAg and STAg in JCPyV can be ascertained from studies of SV40. In SV40, LTAg inhibits the cell cycle via interactions with the tumor suppressor proteins retinoblastoma-associated protein (RB) and p53. SV40 LTAg also exhibits helicase activity, unwinding viral DNA and subsequently facilitating DNA replication [52]. The STAg in JCPyV also contributes to cell cycle arrest by binding to RB proteins as well as protein phosphatase 2A, a significant regulator of the cell cycle [53]. Replication is supported by the three T antigen splice variants, without which JCPyV replication efficiency is reduced [54]. As the virus appropriates the host cell replication machinery to replicate its genome, the late region is transcribed, producing new viral capsid proteins that eventually enclose newly produced viral genomes. Late region transcription of JCPyV also produces agnoprotein. While its function is not fully known, agnoprotein contains a mitochondrial targeting sequence, localizing to mitochondria and altering metabolic functions as a result [55].

## 3. How Is JCPyV Transmitted?

JCPyV can be transmitted via the respiratory route. The respiratory tract expresses an abundance of α2,6-linked sialic acid binding receptors [56]. JCPyV DNA has also been found in the tonsillar tissue of individuals [12]. The tropism of JCPyV for respiratory tract tissue can be examined through the lens of cancer. While many studies have found various polyomaviruses in multiple oropharyngeal and lung cancers, only one study has detected JCPyV, albeit infrequently, in adenoid cystic carcinoma, an aggressive cancer that commonly affects the salivary glands [57,58,59,60,61,62,63]. Moreover, JCPyV virus-like particles (VLPs) carrying miRNA therapies were successfully used to treat lung adenocarcinoma growth and metastasis in vitro and in mice, demonstrating the possibility of transferring genetic material via JCPyV VLPs to the lungs [64]. Similarly, MuPyV is able to cause systemic infection via respiratory inoculation [16,65]. MuPyV VLPs were also used as a vaccine vector against group A Streptococcus, which was delivered intranasally [66]. More work is needed in vivo to elucidate the relationship between JCPyV and respiratory transmission mechanisms [58,67].

JCPyV may also be transmitted via the fecal–oral route. JCPyV infection rates within populations have been monitored using community sewage surveillance, as the virus is readily found in urine and stool [9,68,69,70,71]. During the coronavirus disease 2019 (COVID-19) pandemic, social and contact isolation protocols have decreased the transmission of respiratory-borne illnesses, including rhinovirus, influenza, and respiratory syncytial virus [72]. JCPyV seropositivity rates did not follow this trend and remained consistent instead, regardless of isolation measures [73]. On the other hand, another study, which examined JCPyV seropositivity rates in multiple sclerosis (MS) patients during COVID-19, discovered that conversion rates declined during lock down [74]. Other studies have found an increasing prevalence of JCPyV LTAg in colorectal cancers [75,76,77,78,79]. Our laboratory has demonstrated that MuPyV can cause systemic infection after oral gavage inoculation (AE Lukacher, unpublished observations). MuPyV is suggested to be transmitted through the fecal–oral route as well, evidenced by its shedding in the urine and its availability in drinking water [80,81]. Regardless of the route of infection, JCPyV and MuPyV can both reside in the kidney and other peripheral tissues. MuPyV can be used as a model for JCPyV. Utilizing MuPyV in mice would be key to understanding how polyomaviruses are spread and are able to cause systemic infection.

### Polyomaviruses in Peripheral Tissues

The kidney is a major reservoir for JCPyV infection. Outside the kidney, JCPyV persists in various other tissues, including the spleen, lymph node, lung, and bone marrow [82,83,84]. Since the majority of studies utilize kidney tubules, we will mainly discuss peripheral polyomavirus infection in the kidney. Various studies have diagnosed persistent JCPyV infection via viruria in about 19–45% of healthy seropositive adults [85,86,87,88]. Moreover, a study that examined JCPyV VLP migration in mice found that mice injected with JCPyV VLPs through the tail vein demonstrated the accumulation of VLPs largely in the kidney and the liver [89]. Because viruria is common in asymptomatic carriers and JCPyV is known to attach to and infect kidney epithelium, we can conclude that the kidney is a significant reservoir of JCPyV persistence. Little is known about JCPyV’s life cycle in the kidney. Not only is the kidney a major site of persistent JCPyV infection, but it is also a source of graft failure in kidney transplant patients. Case reports have demonstrated that, while rare, transplant patients can experience JCPyV-associated nephropathy (JC-PVAN) [90,91,92,93,94,95]. The mechanism behind how JCPyV infiltrates the kidney and how JC-PVAN may develop is unknown.

BK polyomavirus (BKPyV), a non-neurotropic human polyomavirus, is a major culprit of BKPyV-associated nephropathy (BK-VAN) in renal transplant patients [96,97]. BK-VAN results in 1 to 10% of graft losses within 12 months [98,99,100,101]. Because this disease is a major cause of infectious graft loss, many studies focusing on BK-VAN have been conducted and could shed light on JCPyV affinity and persistence in the kidney. BKPyV shares about 70% of sequence homology with JCPyV [102,103]. Like JCPyV, BKPyV has a seroprevalence of 80–90% in adults and is similarly transmitted via respiratory or fecal–oral routes [104,105,106,107]. The mechanism behind BKPyV infection and latency in the kidney has been proposed and discussed [108,109]. The kidney tubules were found to be the most permissive to BKPyV replication. The tubules facilitate polyomavirus replication via increased glycolysis and constitutive the expression of transcription factors that steer cells into S phase [110]. Kidney tubules are also considered to be immune-privileged, allowing the virus to replicate freely undetected by the innate or adaptive immune system. Evading surveillance would allow polyomaviruses to replicate in the kidney tubule and be shed in the urine, which is seen for both JCPyV and BKPyV. The mechanism that results in asymptomatic shedding or in the development of end-stage renal disease (ESRD) is unknown and is limited due to the lack of a proper animal model. 

Similarly, MuPyV resides persistently in the kidneys of infected mice [25,111,112,113,114]. Intracranial (IC) infection of MuPyV results in persistent viral DNA in the kidney [115]. How the virus enters the glomerulus, infects the tubules, and establishes persistent infection is yet to be unraveled. Pinpointing the mechanism would be key to developing new ways to diagnose and treat transplant patients at risk of graft-loss due to PVAN. MuPyV, however, has been used in a murine kidney transplant model [111]. Major conclusions from this study reveal that the dominant mechanism of transplant injury in PVAN is due to both host-immunity and virus-induced inflammation, as each alone was insufficient in resulting in permanent graft loss. More studies in vivo need to be conducted to understand the pathophysiology of PVAN. 

## 4. How Do Polyomaviruses Maintain Long-Term Persistence?

In an immunocompetent host, the transmission of JCPyV ultimately results in a persistent infection that is likely kept in check by the immune system, given that JCPyV does not exhibit true latency. JCPyV and MuPyV undergo active lytic infection in vitro, whereas in vivo we do not see signs of viremia unless the hosts are immunocompromised. Interestingly, the literature demonstrates high rates of JCPyV-specific antibodies in the blood [6,116,117,118]. How the virus elicits this strong humoral response is unknown. To elicit a lifelong response, JCPyV and MuPyV may establish a series of lytic infections that continuously prime the immune system; MuPyV viruria appears persistently and episodically [81]. One study demonstrates that MuPyV utilizes its miRNA to promote viruria during acute and persistent infection [81]. Virally encoded miRNAs have been cited for polyomaviruses as a means to both establish persistence and evade host-immunity. MuPyV persistence is also known to be partly driven by enhancers in kidneys. Alterations in these enhancers are shown to diminish both acute and persistent MuPyV infection [119]. Alternatively, the immune system maintains a subclinical smoldering infection. In an immunocompetent host, B cells and T cells control lytic infection because deficiencies in either B cells or T cells allow MuPyV to cause disease [14,25,114,120,121,122,123,124].

Whether persistent infection is elicited by many episodes of lysis, a controlled smoldering infection, or some combination is yet to be answered. Answering these questions is paramount to uncovering the role of lifelong polyomavirus infection and to the development of diagnostics and potential therapeutics for patients at risk of developing PML. 

## 5. How Does Polyomavirus Travel from the Periphery to the Brain?

Although JCPyV infects the kidney, the brain is typically the site of pathology, so the virus must travel between the organs. In patients with PML, the JCPyV in the brain is genotypically different from the virus in the kidney [21,27,125,126,127,128,129,130]. Polyomaviruses residing in the kidney typically exhibit the archetype strain. When patients are immunosuppressed, viral replication is amplified and results in mutations in the NCCR or VP1 [131,132,133]. Viral shedding increases and results in lytic infection. Patient studies have shown that the NCCR undergoes rearrangements in two human immunodeficiency virus (HIV)-positive patients with PML [127,134,135]. One study found that HIV-negative MS patients with PML did not have complex rearrangements, suggesting that either T helper cell depletion or HIV itself may play a role in NCCR rearrangement [29].

Specific mutations in the capsid protein VP1 are also associated with shifts in tropism and can promote neurovirulence and brain infection. JCPyV variants found in the cerebrospinal fluid (CSF) of PML patients exhibit VP1 mutations [136,137]. In mice, VP1 mutations in MuPyV emerge after serially passaging MuPyV in vitro and in the presence of a VP1-specific neutralizing monoclonal antibody (mAB) [25]. Building on this study, our recent work demonstrates that in an immunocompromised setting, MuPyV-infected mice developed MuPyV variants with VP1 mutations, with varying effects on tissue tropism. Mice intracranially infected with a double-mutant VP1^+^ virus displayed more infected cells in the brain. Double mutant virus-infected mice also developed hydrocephalus to a higher degree, indicative of increased neurovirulence of the VP1 double-mutant virus [114]. Taken together, the virus is likely only able to travel to the brain after immunosuppression, which promotes conditions in which the NCCR and VP1 are mutated. This requirement is substantiated by previous studies that show that increasing immunosuppression is directly correlated with increased JCPyV DNA in the brain in individuals without PML [138,139,140,141]. 

Contrary evidence demonstrates that during the autopsy of healthy individuals and in the sampling of pediatric brain cancers, JCPyV resides in the brain with no evidence of brain pathology [126,142,143,144,145,146,147,148,149,150,151,152,153,154]. In mice, peripheral infection with MuPyV does not lead to detectable brain infections. During immunosuppression, MuPyV follows similar trends to JCPyV where viral replication, serum presence, and viruria are all increased [25,81,114]. Further studies need to examine how MuPyV is able to travel from the kidney to the brain to shed light on JCPyV pathogenesis. 

## 6. How Does Polyomavirus Invade the Brain?

To enter and infect the brain, JCPyV must cross one of its multiple barrier sites. The blood–brain barrier (BBB) is the most studied barrier site consisting of vascular endothelial cells, pericytes, and astrocytic end feet [155]. The blood–CSF–brain barrier (BCSFB) exists in the ventricles, as well as on the surface of the brain with separate tissues in each site. Within the ventricles, endothelial and epithelial cell layers of the choroid plexus (ChP) separate the blood from the CSF, whereas ciliated epithelial cells of the ependyma divide the CSF from the parenchyma. On the outer surface of the brain, the meninges are divided into three layers: the dura layer closest to the skull with blood vessels and resident immune cells; the arachnoid layer with permeating CSF; and the pial layer closest to the brain. All three barriers are depicted in Figure 2. The barrier JCPyV uses to establish brain infection is unknown; multiple hypotheses have been proposed, which are detailed below.

### 6.1. Via the Blood–Brain Barrier?

One potential entry site for polyomaviruses into the CNS is the BBB. The BBB exists throughout the brain, dividing the blood vessels from the surrounding parenchyma. The vascular endothelial cells are interconnected by tight junctions and adherens junctions, restricting pathogens from crossing [155,156]. Pericytes, astrocytes, and neurons complement the endothelial cells, sensing and regulating blood flow, thereby forming the overall neurovascular unit. Under homeostasis, few immune cells are permitted to cross this barrier; however, during neuroinflammation, additional immune cells are able to enter.

One proposed mechanism of initial CNS infection by JCPyV is the direct infection of the neuroendothelial cells of the BBB. Cell-free JCPyV has been detected in plasma [157,158]. In culture, JCPyV productively infects primary human brain microvascular endothelial cells [159]. JCPyV has also been shown to bind to sialic acid receptors on the surface of endothelial cells in human brain tissue sections [160]. These same cells lack the expression of the 5-HT_2_ receptors necessary for viral entry. Several studies have described neuroendothelial cells expressing JCPyV transcripts in patient postmortem tissue, as detected by in situ hybridization [161,162]. It is unclear whether this infection occurred from the luminal or parenchymal side of the cells. Given the infrequency of infected neuroendothelial cells and the lack of suitable receptors for binding and internalization, direct infection of the BBB remains an unlikely mechanism for viral entry to the CNS.

Traversal of infected immune cells via the BBB into the brain has been proposed as a “Trojan horse” model of the initial CNS infection. Primary B lymphocytes and CD34^+^ hematopoietic progenitor cells in culture are susceptible to JCPyV infection [163,164], whereas CD34^+^ progenitors and B cells from PML patient spleen [82], bone marrow, blood [157,158,163,165,166,167,168], and brains [169] were positive for JCPyV DNA. However, the virus does not appear to replicate well within B cells or CD34^+^ cells in vitro; the virus bound to B cells mostly remains at the cell surface [164] and JCPyV in primary B cells or B cell lines results in an infection rate under 5% [163,164,170,171]. This low infection rate also occurs in vivo, as few HIV+ or PML patients have viral mRNA in their blood even when JCPyV DNA is detectable [158]. The limited infection rate does not necessarily preclude CD34^+^ progenitor cell or B cell-mediated entry, as a virus attached to or within B cells is sufficient to infect glial cells in the culture [171]. The more recent development of PML in patients receiving monoclonal antibody therapy also complicates JCPyV entry via B lymphocytes. Rituximab is a chimeric mouse–human CD20 antibody that binds and depletes B cells [172] and is used to treat non-Hodgkin’s lymphoma, chronic lymphocytic leukemia, rheumatoid arthritis, and pemphigus vulgaris [173]. Since its approval by the FDA in 1997, other CD20-targeting antibodies have been introduced to the market, including murine, human, and humanized forms [174]. Because these antibodies deplete B cells, and therefore would limit lymphocyte entry into the CNS, it is unclear whether a sufficient number of cells persist during treatment to allow for establishment of brain infection by JCPyV. However, following the cessation of anti-CD20 therapy, CD34^+^ progenitors, immature and naïve B cell populations expand between 6 and 10 months post-treatment, whereas memory B cells remained reduced [175,176]. The expansion of B cells from JCPyV-infected bone marrow may then lead to the dissemination of the virus to the brain after CD20 depletion.

### 6.2. Via the ChP-CSF-Ependyma Barriers?

Another proposed entry site of polyomaviruses is the BCSFB within the ventricles. The ChP contains fenestrated vascular endothelium unlike the BBB, and a stromal layer between the endothelium and the epithelium. The epithelial cells form a barrier with tight and adherens junctions preventing pathogens from entering the CSF. The ependymal cells lining the ventricles are less tightly connected than the ChP epithelial cells, joined by adherens junctions and gap junctions without any tight junctions.

As with the BBB, one potential mechanism of brain entry is the direct infection of the ChP, either of the blood vessels or of the epithelial cells via the fenestrated endothelium. The ChP epithelium and blood vessels express the receptors necessary for JCPyV to bind [160] and JCPyV can productively infect cultured ChP epithelial cells in vitro [41,154] in vivo [177,178]. A particularly interesting feature of ChP infection is the production of JCPyV packaged within EVs [41], which eliminates the need for specific receptors for viral entry. As mentioned earlier, recent studies found that EVs are involved in transmitting the virus to cells independent of 5-HT_2_ receptors [179]. JCPyV-infected ChP cells were able to release EVs containing the virus, which were capable of infecting glial cells [41]. EVs with JCPyV DNA have also been found in human plasma samples [130]. JCPyV-containing EVs have been shown to carry JCPyV-associated DNA, miRNA, and whole viral particles [180]. These data suggest a route of infection between permissive and nonpermissive cell types.

Once the ChP is infected, the virus can be secreted into the CSF. JCPyV DNA has been detected in the CSF of PML patients [168,181] and patients with JCV-associated meningitis [177,182] and is included in the diagnostic criteria for PML [181], although the predictive value of positive or negative PCR results is debated [183,184,185]. Productive MuPyV infection occurs in the ependyma lining the ventricles following intracerebroventricular infection [25,114,124]. In addition to direct infection of the BCSFB, this route of infection may also occur via infected leukocytes, particularly CD34^+^ hematopoietic progenitor cells [186]. As in the blood, anti-CD20 therapy depletes B cells in the CSF, and the cessation of antibody treatment leads to an expansion of CD34^+^, pre- and immature B cells [187].

### 6.3. Via the Meninges?

A third potential entry site for JCPyV is through the meninges on the surface of the brain. The dura mater contains fenestrated vasculature similar to the ChP and also contains lymphatic vessels that drain to the cervical lymph nodes. Below the dura mater, the arachnoid mater is interconnected by tight junctions, restricting pathogens to the dura. CSF flows in the subarachnoid space between the arachnoid and pial layers, or collectively the leptomeninges, and therefore, pathogens already in the CSF can access these layers. In contrast to the fenestrated dural vessels, subpial vessels are similar to the BBB, with tight and adherens junctions preventing pathogens from entering the meninges. The pia mater is more permeable to CSF and allows for the exchange of interstitial fluid in the brain and CSF.

Polyomaviruses may enter first at the dura. Recent literature has found that the dural meninges are a site of infection for *Borrelia burgdorferi* in Lyme disease [188]. The presence of polyomaviruses in the dura has never been studied. Interestingly, B cells within the dura are replenished by the bone marrow within the calvaria of the skull [189,190]; if this bone marrow is also a site of persistent JCPyV, then infected B cells may cross directly from the calvaria to the dura. Polyomaviruses within the CSF can access the leptomeningeal cells within the arachnoid and pial layers. Leptomeninges are susceptible to JCPyV [154,160,177,178] and MuPyV [25,124] infection in vivo and in vitro. The virus may also enter via direct infection of the subpial vessels; JCPyV has been shown to bind sialic receptors on the surface of endothelial cells within the meninges in tissue sections [160].

## 7. What Brain Cells Are Infected by Neurotrophic Polyomaviruses, and What Pathologies Does This Infection Cause?

The pathology of brain polyomavirus infection is driven in part by the specific cells infected. JCPyV and MuPyV virions bind to sialylated gangliosides and serotonergic 5-HT_2_ receptors to be internalized via endocytosis. Polyomaviruses can bypass this requirement, however, by releasing the virus within EVs [41]. These vesicles can then fuse with cells, allowing the virus to enter. Although JCPyV brain infection is associated with PML, the virus can also cause meningitis [124,177,182], granule cell neuronopathy [191,192,193,194], and encephalopathy [195,196]. The presentation of the illness is largely determined by the type of cells infected, which is driven in part by the aforementioned NCCR and VP1 mutations.

### 7.1. Oligodendrocytes

Oligodendrocytes are the myelinating cells of the CNS, providing insulation for the saltatory conduction of axons and trophic support for the underlying neurons. The death of these cells results in loss of neuronal conductance, eventually leading to the development of white matter lesions. In PML, oligodendrocytes are absent from the center of lesions, and at the periphery, infected cells display enlarged nuclei dense with replicating JCPyV [197,198]. The lytic infection of oligodendrocytes causes demyelination of the axons and ultimately many of the symptoms of PML. Mature human oligodendrocytes as well as their precursors do not express the sialic acid LSTc but do express 5-HT_2_ receptors; in tissue sections, JCPyV is not able to bind these cells [160]. This contradicts abundant examples of polyomavirus-infected oligodendrocytes in patient tissue [161,199]. Similarly, MuPyV does not infect cultured oligodendrocytes but is seen within infected cells from mice [120,200]. These conflicting results can be explained by infection via EVs, likely released from the infected ChP [41]. This phenomenon is further exemplified in nude mouse experiments where the route of infection altered the pathology. Human tumor-implanted nude mice showed demyelination and paralysis as a result of MuPyV infection [201,202] while bypassing the meninges and ChP via stereotaxic infections in the striatum resulted in reduced inflammation and demyelination [203]. Human gliachimeric mice [204,205] infected with JCPyV in the corpus callosum developed focal demyelination, although fewer oligodendrocytes were LTAg^+^ or VP1^+^ than other cell types [206]. While this may be due to astrocytic cell death or forced cell cycle entry by the infected oligodendrocytes, an additional explanation is the direct injection of the virus into the corpus callosum limited oligodendrocyte infection by EVs.

### 7.2. Astrocytes

Astrocytes support the functions of many of the other cells in the brain, including sequestering excess neurotransmitter at synapses, taking up glucose from blood vessels and processing it to lactate for neuronal use, and regulating myelination by oligodendrocytes via glutamate signaling [207,208]. These cells also interact with many immune cells, particularly with microglia, the CNS resident myeloid cells, and peripheral immune cells after infection. Such interactions can induce astrocyte reactivity [209,210]. The loss of astrocytes via viral infection disrupts many other cells and can cause neuronal cell death via loss of trophic support. Like oligodendrocytes, astrocytes lack sialic acid receptors and cannot bind JCPyV in tissue sections [160] but express viral proteins in postmortem tissue [161,197]. These cells are also likely infected by EVs [41]. Lesions within PML brains contain “bizarre astrocytes”, cells with enlarged nuclei and stellate processes which are often multinucleated with apparent mitotic spindles [198]. Although oligodendrocytes often undergo lytic cell death after JCPyV infection with virions present in their nuclei, astrocytes typically express LTAg in their cytoplasm with little viral protein in their nuclei [161,198]. In the JCPyV infection of astrocytes in vitro, normal human astrocytes showed the delayed production of VP1 compared to SV40-transformed SVG-A cells [211], suggesting a cell-intrinsic delay in viral replication. In particular, the bizarre astrocytes present in lesions produce little virus but instead have been transformed by viral infection [197,212]. The conserved LTAg in polyomaviruses has been shown to transform and immortalize cells via its interactions with RB [213,214] and p53 [215,216], preventing checkpoint inhibition to allow for continuous divisions. It is not known why viral infection leads to the formation of these bizarre astrocytes rather than immortalization and tumor formation as may be suggested by in vitro experiments, and how these bizarre astrocytes function relative to healthy cells. 

### 7.3. Microglia

Microglia are the resident myeloid cells of the CNS that can phagocytose cellular debris and secrete immune regulators. Under homeostasis, microglia are quiescent and surveil the brain environment. Following an injury or infection, microglia become reactive, migrating, and proliferating while recruiting peripheral immune cells [217]. These cells lack the sialic acid receptors necessary for viral binding [160]. Although JCPyV does not infect microglia, the transcription of MuPyV has been seen in microglia purified from infected mice [120]. Direct infection of the microglia could lead to defects in clearance and immune signaling, impairing the brain’s ability to limit the virus. 

### 7.4. Glial Progenitor Cells

Neural progenitor cells (NPCs) are multipotential stem cells that can differentiate into astrocytes, oligodendrocytes, or neurons, and are important to replace damaged cells after an injury. JCPyV can non-productively infect NPCs in vitro [218], although direct infection of these cells or a bipotential glial precursor line CG-4 results in a reduced ability for the precursors to differentiate [218,219]. Under homeostatic conditions, oligodendrocyte precursor cells can migrate and differentiate to replace damaged oligodendrocytes to maintain myelin. Mice continue to undergo myelination after the first 6 weeks [220] in humanized glia/mice, the murine oligodendrocytes are replaced by human cells over time [205]. Humanized glia/mice 12 weeks post-infection with JCPyV showed hypomyelination in the corpus callosum and internal capsule compared to uninfected chimeras [206], suggesting a defect in the ability of precursors to differentiate and myelinate. In an inflammatory MS model, oligodendrocyte precursor cells exposed to interferon-γ (IFN-γ) failed to differentiate to mature, myelin-producing cells and instead presented peptide on major histocompatibility complex (MHC) class I and were eliminated by CD8^+^ T cells [221]. It is possible the inflammatory conditions from JCPyV infection also limit the differentiation of precursor cells, thereby limiting remyelination. 

### 7.5. ChP Epithelium and Ependyma

The ChP epithelium is the primary secreter of CSF as well as driving CSF flow throughout the brain via its ciliated surface. As mentioned previously, these cells provide a barrier from the fenestrated blood vessels to the CSF. Either direct infection or the inflammation of ChP epithelial cells can cause hypersecretion of CSF leading to the development of hydrocephalus [222,223,224]. ChP epithelia express both LSTc and 5-HT_2_ receptors [160] and are permissive for JCPyV infection in vitro [41,154] and in vivo [177,178]. Infection of the ChP leads to the development of hydrocephalus in JCPyV meningitis rather than typical symptoms associated with PML. The ChP epithelium expresses multiple pattern recognition receptors (PRRs), including toll-like receptors (TLRs) [222,225,226,227,228] and C-type lectin receptors [229], but the immune sensing of ChP cells during polyomavirus infection has not been explored. Immunodeficient mice intracerebroventricularly infected with MuPyV do not show direct infection of ChP epithelium but do develop severe hydrocephalus [124], suggesting the inflammatory signaling by MuPyV alone is sufficient to lead to CSF overproduction. In addition to altering CSF secretion, infection of the ChP can lead to the downregulation of tight junctions, increasing the permeability of the BCSFB. In particular, inflammation in a stroke mouse model [230], infection of cultured human ChP epithelial cells with *Borrelia burgdorferi* [231], or mice modeling cerebral toxoplasmosis [232] can lead to the reduced expression of tight junction genes and proteins. The effect of polyomavirus infection on ChP epithelial permeability has not been determined.

The ependyma is a ciliated cell layer that lines the ventricles and promotes the flow of CSF. This layer also regulates the exchange of CSF and interstitial fluid within the brain parenchyma and likely produces CSF. Therefore, a defective ependyma barrier also contributes to hydrocephalus and ventriculomegaly [233,234,235,236]. While direct infection of the ependyma by JCPyV has not been seen, MuPyV infects the ependymal lining causing denudation in immunodeficient mice [124]. Inflammation of the ependyma via stimulation of its PRRs also leads to hydrocephalus and ventriculomegaly [225,228]. As with the ChP, inflammatory signaling at the ependyma may contribute to the pathology seen with JCPyV meningitis, but this has not been investigated. 

### 7.6. Leptomeningeal Cells

The leptomeninges include the arachnoid and pial meninges layers surrounding the exterior surface of the brain. The arachnoid mater regulates CSF exchange to the glymphatic system to be drained, while the pia mater regulates interstitial fluid-CSF exchange. Inflammation and infection of the leptomeninges can disrupt both the flow and clearance of the CSF [237,238] and can lead to hydrocephalus in addition to the development of meningitis [239,240]. Leptomeninges are productively infected by JCPyV in vitro [160,178,241] and in vivo [177,178,241], leading to JCPyV meningitis without the symptoms of PML. Mouse meningeal infection with MuPyV in vivo and in vitro replicates this finding [25,124]. Considering the importance of the meninges to the clearance of the CSF, determining the potential disruption caused by polyomavirus infection could help with understanding pathology in JCPyV meningitis and PML.

### 7.7. Neurons

The neurons are electrochemically coupled cells that carry messages along their axons to neighboring neurons. Different regions of the brain correspond to various functions, and therefore the location of infection in or around neurons will determine its impact. One subset of neurons, the granule cells of the cerebellum, receive sensory inputs from the body and project its axons to the molecular layer [242]. These cells are susceptible to JCPyV infection both with PML [243,244] and alone, causing JCPyV granule cell neuronopathy [191,192,193,194]. With neuronal cell death, granule cell neuronopathy causes cerebellar dysfunction with symptoms like gait ataxia, loss of coordination, and dysarthria. Specific mutations in the C-terminus of VP1 are often associated with granule cell infection that reduces the ability of the virus to infect glia [245,246]. JCPyV can also infect cortical pyramidal neurons, leading to the development of JCPyV encephalopathy [195,196]. Patients with infected cortices develop cognitive decline and aphasia.

## 8. What Is the Innate Immune Response to Polyomavirus?

All viruses are obligate intracellular pathogens. The antiviral immune response thus involves a series of evolved mechanisms that not only dampen viral replication but also signal the infection to neighboring cells and recruit the innate and adaptive immune system. While some of the signaling mechanisms for the innate immune response have been determined, many remain unstudied.

### 8.1. Sensing, Reporting, and Alarming Innate Immunity

Antiviral immunity can be divided into three systems: sensing, reporting, and alarming. For polyomaviruses, the sensing system is an active field of research with many questions unanswered. Sensing is performed by PRRs, a diverse group of receptors that sense either extracellular or intracellular molecules associated with pathogens. On the extracellular front, TLRs, especially TLR4 and TLR9, are important in sensing neuropathic polyomaviruses like JCPyV and MuPyV [247,248,249,250]. One JCPyV vaccine study has looked into the interaction of TLR4 with JCPyV VP1 capsid protein as a mechanism of priming immunity [249]. Single nucleotide polymorphisms in TLR3 and 4 contribute to BKPyV viremia in transplant patients [251,252]. Although TLR sensing of MuPyV has not been extensively studied, one study demonstrated a direct interaction between MuPyV VP1 and murine TLR4 in primary and untransformed mouse embryonic fibroblasts and showed that TLR4-mediated recognition of the virus led to cancer-like phenotypes [247]. 

Intracellularly, PRRs sense both DNA and RNA viruses in the cytoplasm. In a healthy host cell, DNA should be localized within either the nucleus or the mitochondria. The PRR cyclic GMP–AMP synthase (cGAS), a double-stranded DNA sensor, appears to be a key sensor of polyomavirus [253,254]. cGAS drives interferon-β (IFN-β) production during MuPyV infection in vitro when genotoxic stress is achieved in murine fibroblasts [253]. Early in its replication cycle, however, MuPyV fails to induce IFN-β due to lack of interaction with cGAS until the virus undergoes replication in the nucleus.

Another study suggests that JCPyV T antigens do not interact with the cGAS sensing pathway [254]. Retinoic-acid inducible gene I (RIG-I), an RNA-sensing PRR, has been implicated in the role of JCPyV infection of SVG-A human cell lines [254]. JCPyV STAg interacts with RIG-I directly to suppress its signaling. Outside of these two studies, not much is known about how neurotropic polyomaviruses interact with antiviral sensors and downstream effectors. One study has found that endothelial cells are the first line of defense against BKPyV infection and can release low levels of IFNs [255]. An additional study found that double-stranded RNA sensors, like TLR3, RIG-I, and melanoma differentiation-associated protein 5 (MDA5), all play a role in BKPyV infection [256]. Studies have not been conducted for JCPyV and MuPyV and present a gap in the field that needs to be addressed. Understanding the antiviral and innate immune system responses is essential to understanding how to resolve persistent infections at the cellular level and develop novel therapeutics to utilize these systems against polyomavirus diseases.

Sensing mechanisms target signaling mechanisms to release respective alarms, also known as IFNs. There are three types of IFNs, conveniently named type I, type II, and type III. All cells express type I IFNs via PRRs and reporter pathways, including IFN-α and IFN-β. Type II IFN (IFN-γ) is derived from natural killer (NK) cells and cytotoxic T cells. Type III IFNs cover all the IFN-λ subtypes that are dominantly expressed by epithelial, BBB endothelial cells, macrophages, neutrophils, and subsets of dendritic cells [257]. All the DNA and RNA sensing mechanisms mentioned stimulate the expression of type I IFNs. Interferon regulatory factor 3 (IRF3) is essential for the expression of IFN-β [253,258]. DNA sensing via cGAS stimulates the production of cyclic GMP-AMP (cGAMP), which binds to its adapter, stimulator of interferon genes (STING), leading to activation of IRF3 to initiate the transcription of IFN-β. RNA sensing via RIG-I and MDA5, another PRR, initiates binding with their adapter, mitochondrial antiviral-signaling protein (MAVS), which also leads to IRF3 activation. Additionally, TLR4 is also known to activate IRF3, whereas TLR9 stimulates the phosphorylation of IRF7, helping initiate the transcription of IFN-α [259,260]. Downstream adapters have been implicated but not adequately studied in polyomaviruses. As previously discussed, MuPyV interacts with cGAS and p204 during infection of murine fibroblasts, also activating IRF3 and stimulating IFN-β production [253]. Another study found that IFN-α and IFN-β are both negative transcriptional regulators of JCPyV infection in human glial cells via the induction of CAAT/enhancer binding protein beta (C/EBPβ) and liver inhibitory protein (LIP) [261]. The downregulation of C/EBPβ-LIP via shRNA treatment caused the reactivation of JCPyV. The field of polyomaviruses and antiviral immunity is steadily growing, and studies have not articulated which sensing-adaptor pathways prevail and what their function might be within a more complex host-model. More studies need to be conducted, especially in the context of neurovirulence, as this pathway is crucial in intracellular reporting of viral activity, which could play unique targets in treatment and therapeutics. 

Janus kinase (JAK) proteins and signal transducer and activator of transcription (STAT) transduce signals from all IFN receptors. Gain-of-function mutations in STAT1 have been associated with the development of PML [262]. One case report has linked JAK-STAT inhibitors with the incidence of fatal encephalopathy due to JCPyV [195], so understanding the effects of JAK-STATs in the context of neurovirulent polyomavirus infection is imperative to understanding the development of PML. In patient samples, JCPyV infectivity increased in renal tubules during type I IFN blockade [263]. Activated STAT1 and IRF9 were found to initiate transcription of interferon-stimulated genes (ISGs), which were thought to dampen JCPyV replication in primary renal proximal tubular epithelial cells and establish latency. Without a mouse model, conducting these studies in vivo presents a challenge.

MuPyV has provided an avenue to study the effects of types I and II IFNs against MuPyV-induced encephalitis. Similar to what has been shown regarding IFN regulation of JCPyV, MuPyV has been found to be regulated by type I IFN in vitro and played a role in the recruitment of cytotoxic CD8^+^ T cells in vivo. Transgenic mice lacking the interferon-α/β receptor (IFNAR) demonstrated increased viral titers in vivo but also demonstrated increased proliferation, functionality, and memory differentiation of CD8^+^ cells [121]. Additional studies with MuPyV show that type II IFNs increase viral load and the presence of MuPyV-induced tumors in vivo [264]. These pathways both point to the importance of STAT1 within the infected cell. During MuPyV intracerebroventricular infection, STAT1 and CD8^+^ T cells work in concert to dampen neurovirulent polyomavirus load and ameliorate neurodegeneration [124]. STAT1 null mice infected with MuPyV demonstrated increased brain viral loads compared with type I and type II IFN receptor-deficient and WT mice infected with MuPyV. Additionally, MuPyV-infected STAT1-null mice developed severe ventriculomegaly and elevated CD8^+^ T cell infiltrates. *Stat1* null mice depleted of CD8^+^ T cells die within two weeks of infection. Because the ependyma is both directly infected with MuPyV [124] and can be inflamed indirectly via the activation of TLRs [225,228], loss of STAT1 specifically at this barrier leads to increased neuroinflammation after MuPyV infection (SA Spencer and AE Lukacher, unpublished observations).

It is unclear which innate antiviral pathways are protective or lead to encephalopathy. For example, STAT1 null mice show neuropathology, but STAT1 gain-of-function mutations are associated with the development of PML. More studies need to be conducted to elucidate neuroprotective versus neuropathogenic mechanisms. 

### 8.2. The Innate Immune Enzyme Family APOBEC3: A Driver of Polyomavirus VP1 Mutations?

Apolipoprotein B mRNA-editing catalytic polypeptide-like 3 (APOBEC3) genes are antiviral ISGs that code for a family of cytidine deaminases [265,266]. APOBEC3s act on single-stranded DNA, inducing mutagenic damage by converting cytosine to uracil [267]. They are well-studied in the context of retrovirus restriction, with one member having initially been identified as an inhibitor of HIV-1 infection [268]. Due to their influence on viral genomes, APOBEC3s are thought to play a role in the evolution of multiple viruses, including polyomaviruses. For instance, the genomes of JCPyV and other human polyomaviruses, such as BKPyV, KI polyomavirus, and WU polyomavirus all show footprints of APOBEC3 activity [269]. APOBEC3s have particularly been implicated in causing BKPyV mutations that may bestow a selective advantage to the virus. In settings of immunosuppression, BKPyV can abundantly replicate and cause nephropathy, a scenario that occurs in 1 to 10% of kidney transplant patients [106]. Observed VP1 mutations in BKPyV variants isolated from nephropathy patients were consistent with APOBEC3-associated damage [270]. Moreover, BKPyV infection of human urothelium results in increased APOBEC3A and APOBEC3B expression [271].

The involvement of APOBEC3s in influencing JCPyV intra-host evolution in the context of pathogenicity has not been studied. As discussed previously, changes in VP1 can alter viral tropism, receptor binding, and recognition by the host’s neutralizing antibody response [137,272]. Similar to what was observed in BKPyV, APOBEC3s may not actually control JCPyV, instead presenting a driver for VP1 mutations that inadvertently confer an advantage to the virus, generating variants that can escape the host’s neutralizing antibody response. The involvement of APOBEC3s in influencing JCPyV intra-host evolution in the context of pathogenicity has not been studied. As discussed previously, changes in VP1 can alter viral tropism, receptor binding, and recognition by the host’s neutralizing antibody response [137,272]. Similar to what was observed in BKPyV, APOBEC3s may not actually control JCPyV, instead presenting a driver for VP1 mutations that inadvertently confer an advantage to the virus, generating variants that can escape the host’s neutralizing antibody response. MuPyV is a model in which this hypothesis can be tested, with some considerations. Unlike humans, mice have a singular APOBEC3 gene, which encodes an antiviral protein that is largely studied against mouse retroviruses [273]. Interestingly, while mouse APOBEC3 (mA3) is capable of cytosine deamination, it impedes retroviruses via mechanisms that are independent of cytosine deamination, such as the inhibition of reverse transcriptase in murine leukemia virus infection [274]. Essentially, mA3 can restrict murine retroviruses without inducing mutagenic damage. Whether mA3 interacts similarly with MuPyV is unknown. Clarifying the method of attempted restriction is important if using MuPyV to study how APOBEC3 is implicated in driving polyomavirus mutations. 

## 9. What Is the Adaptive Immune Response to Polyomavirus Infection?

Prolonged immunosuppression is a critical risk factor in the development of PML. Historically, PML was associated with acquired immunodeficiency syndrome (AIDS), with 2 to 5% of patients developing PML after AIDS onset [1]. While the incidence of HIV-associated PML has decreased since the implementation of highly active antiretroviral therapy, which manages AIDS through the suppression of HIV at different viral life cycle stages, PML has become a possible complication of relatively recently developed immunomodulatory drugs. For instance, natalizumab, an immunomodulator used to treat MS, inhibits lymphocyte trafficking into the brain with the intent of reducing inflammation [275]. When combined with risk factors such as high JCV seropositivity and prolonged and/or previous use of immunosuppressants, inhibition of lymphocyte trafficking can lead to the development of PML [275]. The adaptive immune response is carried out by cytotoxic CD8^+^ T cells, CD4^+^ T helper cells, and antibody-producing B cells; we will discuss the contribution of each cell type to the control of polyomavirus infection.

### 9.1. CD8^+^ T Cells

The significance of CD8^+^ T cells in polyomavirus control is emphasized by studies demonstrating a strong association between an early JCPyV-specific CD8^+^ T cell response and a favorable clinical outcome in PML patients [276,277]. In clinical studies exploring pembrolizumab, an inhibitor of the immunosuppressive protein programmed cell death protein 1 (PD-1), as a treatment for PML, an increased JCPyV-specific CD8^+^ T cell response was linked with lower viral loads and, ultimately, clinical improvement [3]. CD8^+^ T cells have been shown to colocalize with JCPyV-infected glial cells, further validating the importance of CD8^+^ T cells in controlling JCPyV [278,279]. The necessity of CD8^+^ T cells for viral control in an immunocompromised setting is demonstrated in vivo in STAT1 null mice, which are immunocompromised and exhibit high viral loads in the brain when infected with MuPyV. When depleted of CD8^+^ T cells, these mice display severe weight loss and increased mortality by two weeks post-infection compared to WT animals [124].

CD8^+^ resident memory T (T_RM_) cells constitute a subset of CD8^+^ T cells and are critical responders to viral reinfection in non-lymphoid tissues, such as the brain. T_RM_ cells are semi-permanent fixtures of non-lymphoid tissues and are responsible for rapidly countering repeated antigen encounters [280,281]. The expression of chemokine receptor CXCR6 is associated with the transition of CD8^+^ T cells from circulating effectors to resident memory [282,283]. Previous studies have shown CXCR6 and its chemokine ligand CXCL16 to be important for T_RM_ formation in the lung [284], skin [285,286], and liver [287]. In the brain, T_RM_ cells require CXCL16-CXCR6 signaling after resolution of West Nile virus infection for maintenance, as loss of CXCR6 leads to reduced numbers of CD8^+^ cells [288]. This chemokine signaling was also required for T_RM_ formation in an Alzheimer’s disease mouse model, and CXCR6 null mice had reduced CD8^+^ T cells with increased cognitive decline [289]. The dominant source of CXCL16 in the brain is myeloid cells including microglia [288,289]. With MuPyV infection, we have observed CXCR6^+^ CD8^+^ T cells in the kidney and brains of infected mice, as well as expression of CXCL16 by brain microglia (M Lauver, S Spencer, and A Lukacher, unpublished observations). In CXCR6 deficient mice, CD8^+^ T cells show a reduced expression of T_RM_ marker CD69 [290] after infection, suggesting a requirement of CXCR6 for the establishment of T_RM_ cells. Mice without CXCR6 signaling infected with MuPyV showed a reduced expression of PD-1 on CD8^+^ T cells in the brain (S Spencer and A Lukacher, unpublished observations).

PD-1, a classic inhibitory immune receptor, is an intrinsic attribute of virus-specific CD8^+^ brain-resident T_RM_ (bT_RM_) cells established during MuPyV infection [115]. Notably, PD-1 expression is upregulated on JCPyV-specific CD8^+^ cells in PML patients [289]. Increased PD-1 and PD-L1 expression have also been observed in autopsied PML lesions [3]. MuPyV-specific CD8^+^ bT_RM_ cells highly express PD-1, a characteristic not shared by MuPyV-specific memory CD8^+^ cells in the spleen, supporting PD-1 as a brain-intrinsic trait of CD8^+^ T_RM_ cells [115]. Disrupting PD-1 signaling influences bT_RM_ cell differentiation. MuPyV-infected mice lacking programmed cell death ligand 1 (PD-L1), one of the two canonical ligands for PD-1, present a higher frequency of virus-specific CD103^+^ CD8^+^ cells compared to wild-type mice [115]. CD103 is an alpha integrin considered to be a broad marker of T_RM_ cells [291]. Wild-type mice treated with a blocking PD-L1 antibody also exhibit this increase in CD103^+^ cell frequency, an observation recapitulated by PD-1^fl/fl^ Rosa-Cre^ERT2^ mice, in which PD-1 can be conditionally deleted with tamoxifen administration. Upon PD-1 deletion after MuPyV infection, these mice demonstrate an increase in CD103^+^ cell frequency compared to vehicle-treated animals [120] (A Butic and AE Lukacher, unpublished observations). PD-1 has also been shown to constrain the effector functions of MuPyV-specific CD8^+^ bT_RM_ cells ex vivo. Bone marrow-derived dendritic cells (BMDCs) from wild-type and PD-L1^−/−^ mice were used to stimulate brain CD8^+^ cells isolated from MuPyV-infected mice. PD-L1 null BMDCs incited a lower frequency of cells that were positive for IFN-γ, a classically inflammatory and antiviral cytokine, compared to wild-type BMDCs [120]. Inflammatory gene expression in wild-type and PD-L1 deficient mice indicates that inflammatory genes were upregulated in PD-L1^−/−^ mice during acute infection [120]. Similarly, brain sections from acutely infected, tamoxifen-treated PD-1^fl/fl^ Rosa-Cre^ERT2^ mice exhibit increased glial fibrillary acidic protein (GFAP) and ionized calcium-binding adaptor molecule 1 (Iba1) immunofluorescent staining compared to vehicle-treated mice (A Butic and AE Lukacher, unpublished observations). GFAP and Iba1 are markers of reactive gliosis, an aspect of neuroinflammation. More specifically, GFAP is upregulated in activated astrocytes, whereas Iba1 is increased in microglia [292,293]. Unexpectedly, inflammatory gene expression is reduced in PD-L1 null mice during persistent infection [120]. Collectively, these findings portray PD-1 as a dynamic regulator of MuPyV-induced neuroinflammation and a potentially important factor in MuPyV-specific bT_RM_ cell differentiation. Exploring the factors that contribute to bT_RM_ cell differentiation and function and maintenance is critical to designing therapies for PML as well as improving patient stratification for therapies such as pembrolizumab. Although pembrolizumab has shown clinical effectiveness in some PML patients, its effects are still inconsistent, with some patients showing no response [3,294].

### 9.2. CD4^+^ T Cells

Given that PML is associated with HIV, which classically infects CD4^+^ T cells and thus depletes this crucial cell type over time, the requirement of CD4^+^ T cells in JCPyV control is increasingly appreciated [295]. PML patients who survived more than a year after onset appeared 4.8 times more likely to exhibit a discernable CD4^+^ T cell response compared to PML patients who passed away within a year of onset—an unsurprising observation considering the interplay between CD4^+^ and CD8^+^ T cells [276]. Furthermore, CD4^+^ T cell numbers were found to be inversely correlated with JCPyV DNA levels in the brain, implicating CD4^+^ T cells in the promotion of viral infiltration of the CNS [141]. 

CD4^+^ T cells support several aspects of a robust CD8^+^ T cell response, from initially facilitating antigen presentation by dendritic cells to aiding the expansion of memory CD8^+^ T cells [296]. Mice depleted of CD4^+^ T cells have increased virus levels in the kidney along with higher rates of viremia compared to mice depleted only of CD8^+^ T cells [114]. Viral control in the kidneys may thus be dependent on the help provided by CD4^+^ T cells. During chronic viral infection as modeled by the lymphocytic choriomeningitis virus, CD4^+^ T cells are a critical source of interleukin (IL)-21, without which CD8^+^ T cell responses are severely impaired [297,298]. In MuPyV infection of the CNS, a subset of virus-specific CD4^+^ T cells secrete IL-21, and disrupted IL-21 signaling results in fewer CD8^+^ bT_RM_ cells [299]. As CD4^+^ T cells drive B cell differentiation, they are important for the development of a broad antiviral-neutralizing antibody repertoire. 

### 9.3. B Cells

Despite being potential disseminators of JCPyV, B cells may also be involved in the control of polyomavirus, a role supported by rituximab-associated PML [300]. B cells constitute an important part of the antiviral response, as they are responsible for generating antigen-specific antibodies and are part of immunological memory, which is critical for responding to previously encountered pathogens. In a study following four PML patients, three died within a few months after the onset of PML; the survivor exhibited an increase in neutralizing titers over time in both plasma and CSF, suggesting that the humoral response may be predictive of PML outcome [301]. Moreover, MS patients with PML exhibited a lack of expansion of B and T cells [302]. 

MuPyV infection of B cell-deficient mice leads to chronic viremia and lack of neutralizing antibody response [303]. B cell deficiency with T cell impairment leads to the emergence of antibody-resistant and possibly neurotropic MuPyV mutants [114]. MuPyV-infected B cell-deficient mice were immunized with a VP1 monoclonal antibody to mimic a restricted antibody response, often seen in PML patients that are unable to neutralize JCPyV variants with VP1 mutations [114,304,305]. When these mice were additionally depleted of both CD4^+^ and CD8^+^ T cells, antibody-resistant MuPyV variants with VP1 mutations arose. Overall, a restricted antibody response in concert with T cell deficiency provides the ideal setting for resurgent MuPyV replication and the emergence of mutations in VP1 antibody epitopes. 

How B cells control polyomavirus infection in the CNS remains to be seen. A reservoir of developmentally immature B cells exists in the calvaria bone marrow, presenting a non-systemic source of surveilling B cells [189,190]. Using MuPyV as a model, it would be interesting to explore how this immunological niche may control polyomavirus infection.

## 10. Conclusions

The route that JCPyV follows from the periphery to the CNS remains unanswered. JCPyV enters the periphery likely through the respiratory or fecal–oral route, establishing a persistent infection in the urinary tract and other peripheral sites. The mechanisms behind JCPyV persistence are not fully understood, but it is possible that the virus establishes a low-level infection via undefined means. In an immunocompromised setting, unchecked viral replication can lead to mutations in NCCR and VP1 that shift tissue tropism, allowing mutants to evade humoral immunity and become viremic. Deficiencies in both T and B cells can promote a narrowed neutralizing antibody response, reducing the chances of viral neutralization and giving the virus access to the CNS. It is not clear how the virus enters the CNS, but hypotheses have been proposed for its ability to penetrate each of the three CNS barriers: the BBB, the BCSFB, and the meninges. Infection may occur directly, with the virus infecting endothelial cells, or indirectly, with the virus potentially infecting B cells and CD34^+^ hematopoietic progenitor cells to transit these barriers. As prolonged immunosuppression is a significant factor in JCPyV infection of the CNS, PML is associated with some but not all agents that cause depressed or modulated immunity; thus, it is critical to understand the immune response to polyomavirus infection. Both the innate and adaptive response offer various control mechanisms, from sensing the virus and inducing IFN-I expression to trigger antiviral effector ISGs to a broadly neutralizing antibody repertoire and the formation of the bT_RM_ cells. Exploring polyomavirus CNS pathogenesis and immunity is becoming advanced using the MuPyV model. Studying a neurotropic polyomavirus in its natural host has yielded valuable insight into early CNS pathogenesis and may be the key to improving therapeutic strategies for PML. 

## Figures and Tables

**Figure 1 viruses-15-02112-f001:**
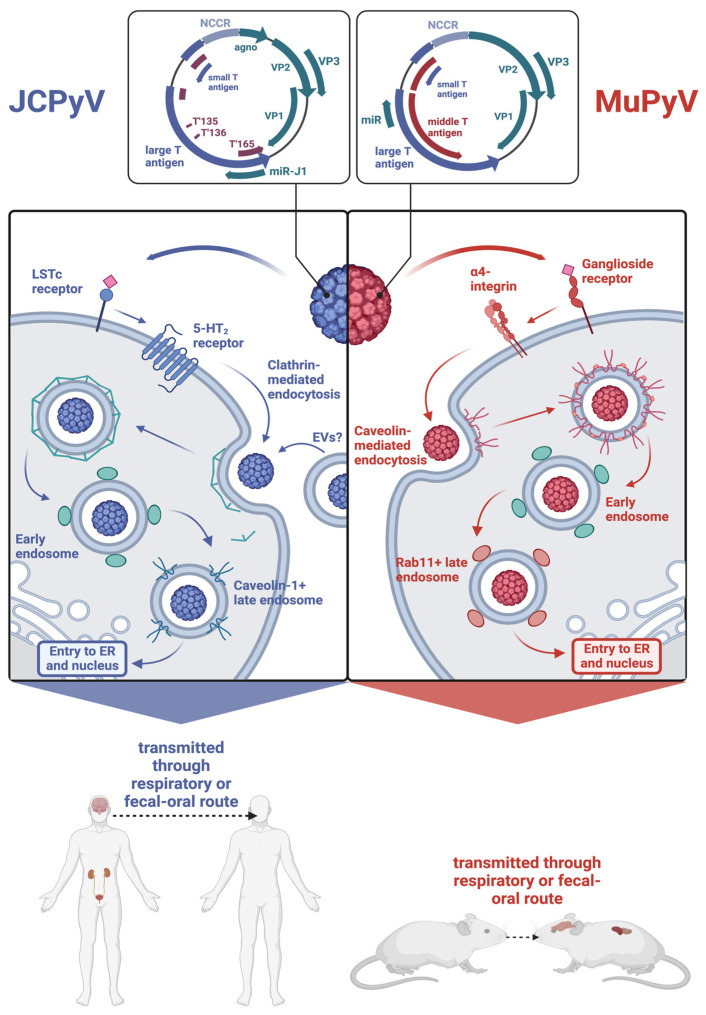
Comparison of JCPyV and MuPyV genomes and how each virus enters its host cell. Both viruses initially attach to a receptor containing a sialic acid. Entrance is then facilitated by a secondary receptor. Both viruses are then internalized via different means. While JCPyV is internalized via clathrin-mediated endocytosis, MuPyV is internalized via caveolin-mediated endocytosis. The viruses then enter the early endosomes and transition into late endosomes. Both MuPyV and JCPyV are transmitted via similar routes and establish infections in similar tissues. Made in BioRender.

**Figure 2 viruses-15-02112-f002:**
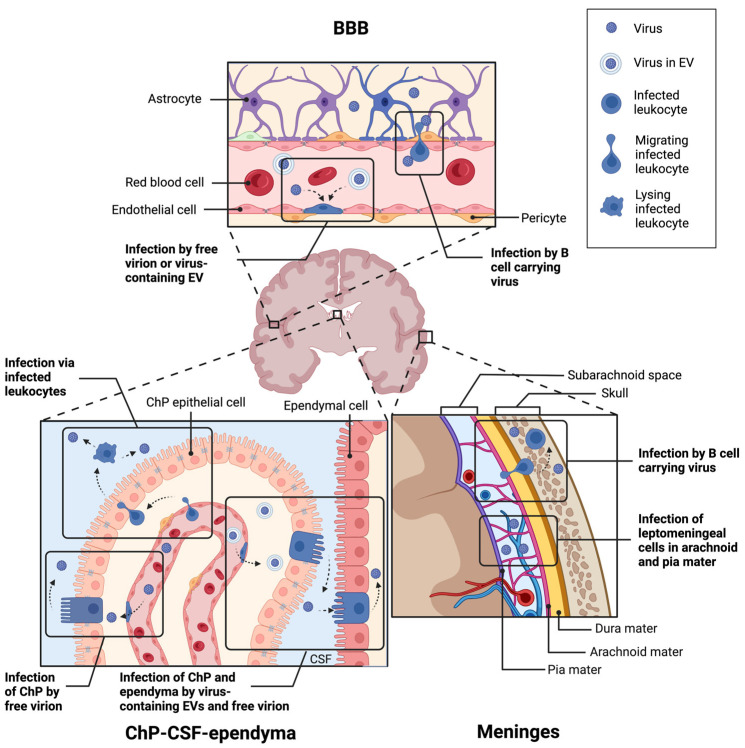
Hypothesized infection routes through the three barriers of the CNS: the BBB, the ChP-CSF-ependyma barrier, and the meninges. Generally, for all three barriers, infection may occur directly via free virions or through indirect means like EVs or infected leukocytes. Made in BioRender.

## Data Availability

Not applicable.

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
