# Peer review of "Polyomavirus Wakes Up and Chooses Neurovirulence"

_viruses, 2023, doi:10.3390/v15102112_

Round 1
Reviewer 1 Report
The review article by Butic, Spencer, Shaheen, and Lukacher is an outstanding and comprehensive review of the challenges and promises of developing an animal model of PML. They compare and contrast murine polyomavirus and JC virus in terms of the molecular biology, the pathophysiology, and the immunobiological mechanisms employed and encountered by each in their respective hosts. This review is contemporary and will serve the field well. Novel insights into the pathogenesis of PML are shared freely and are appreciated.
Minor comments:
Line 241-242. The sentence begins with “Answering these questions are unanswerable in vitro….” Is very awkward and should be re-imagined.
Line 289. I think you mean multiple hypotheses as opposed to theories here.
Line 773. I think you mean hypotheses as opposed to theories here.
Reviewer 2 Report
The document is a well-written review. I make a few suggestions for the authors to consider.
Split the section ‘Structure and lifecycle of polyomaviruses: JCPyV and MuPyV’ to concentrate on the comparison of the viruses and, separately, the life cycle that compares the infectious process. This may better fulfil the stated objective to describe how accurately the murine infection models human pathology.
Suggest other possible comparisons such as phylogenetic comparison among the polyomavirus to show the relatedness between JCPyV and MuPyV.
Suggest further discussion of the infection route, emphasizing unique aspects of murine and human nasal physiology that may affect the effect of the murine model to mimic human disease
I’d suggest a discussion about the possibility of the direct infection of the nervus terminalis neurons or other cranial nerves via the mucus membranes, which connect the olfactory epithelium directly with brain structures. This is in addition to the current assertion that infection proceeds via the meninges.
Related to this, I’d suggest discussing the possibility that infection of the CSN could proceed by reinfection of a host via cranial nerves with a mutant variant virus, rather than via other organs such as the kidney as is currently asserted.
The section ‘Sensing, reporting, and alarming innate immunity’ could be improved to make it more informative. Perhaps this section could be reorganised to distinguish the immune response that suppresses infection and the inflammatory response that causes morbidity in encephalopathy.
Aspects of the current discussion in this immune section are not accurate. Instances include:
Ln 559 - With potentially hundreds of type I IFNs [there are not hundreds of these cytokines]
Ln 560 Type II IFNs include IFN-γ [IFN-γ is the only type II IFN]
Ln 562 Type III IFNs are a recently discovered [these cytokines were discovered 20 years ago]
Ln 562 IFN-λ subtypes that are uniquely expressed by epithelial cells [some immune cells also express these cytokines]
The speculation that APOBEC3s may not actually control JCPyV infection but act as a driver for mutations is an intriguing idea for pathogenicity that could be expanded upon.
Reviewer 3 Report
This review paper is very informative and educational for those interested in polyomavirus (JCPyV) in association with the body or the CNS diseases. I have a few comments about it.
#It was interesting to know that JCPyV or virus of that kind resides throughout the body. According to the authors, these kinds of virus are implicated in a variety of human diseases other than PML, for example, BK-VAN. It would be more wonderful to have a tabulation in this paper, which shows the list of JCPyV-related or polyomavirus-related human diseases so far known or reported, of course including PML, along with the organs infected or the clinical features (ex. Immunocompromised or Immunocompetent).
#7. What brain cells…infection cause?
It was interesting to read this section. In my understanding, the CNS cells that can be infected by JCPyV, i.e., the 6 different cell types in all, are listed up. This is okay and very good, but I just wonder what brain cells are NOT infected by JCPyV. For example, microglia are not present in this list. It would be more wonderful if the brain cells that are not known to be infected by JCPyV, or those that have not been shown to be infected by JCPyV, are documented in this paper.
#Just a bit of editing, which is a must to be published as a review article, is necessary.
(Ex)
-line 70: [17,20].). > [17,20].
-line 120, ‘ER’: What is ER? Rough endoplasmic reticulum? I guess its full-spelling is missing.
-line 127: disrupt > disrupts?
-line 139: Rb > RB?
-line 208: ‘9/30/2023 11:32:00 PM’ should be erased.
-etc, throughout the Ms.
Good.
